# Health Philosophy of Dietitians and Its Implications for Life Satisfaction: An Exploratory Study

**DOI:** 10.3390/bs7040067

**Published:** 2017-10-19

**Authors:** Patricia Grace-Farfaglia, Denise Pickett-Bernard, Andrea White Gorman, Jaleh Dehpahlavan

**Affiliations:** 1Health Science, Rocky Mountain University of Health Sciences, 122 E 1700 S, Provo, UT 84606, USA; agorman@rmuohp.edu; 2Nutritional Sciences, University of Connecticut, 99 E. Main Street, Waterbury, CT 06702, USA; 3Nutrition Department, Life University, 1269 Barclay Cir, Marietta, GA 30060, USA; denise.pickett-Bernard@life.edu (D.P.-B.); Jalehd678@gmail.com (J.D.)

**Keywords:** life satisfaction, healthy lifestyle, registered dietitian nutritionists, health philosophy, integrative practice, gender

## Abstract

Studies of health providers suggest that satisfaction with life is related to their values and sense of purpose which is best achieved when their professional role is in harmony with personal philosophy. Cross-sectional surveys suggest that personal health beliefs and practices of health professionals influence their clinical counseling practices. However, little is known about the influence of health philosophy on the personal satisfaction with life for dietitians. This study recruited a randomly selected, cross-sectional sample to complete a self-administered online survey. An exploratory factor analysis of was conducted for 479 participants resulting in a two-factor solution, clinical (α = 0.914) and wellness (α = 0.894) perceptions of health. An index score for the following valid and reliable scales were calculated: satisfaction with life, health conception, and healthy lifestyle and personal control. Pearson correlation coefficients between scores were analyzed to determine the degree of relationship. Potential mediators were explored with multiple regression. The relationships between variables were tested with structural equation modeling using a multigroup comparison between genders. The male participants were removed from the overall model and were separately evaluated. Health philosophy that is oriented toward wellness, was positively and significantly associated with life satisfaction, *r*(462) = 0.103, *p* < 0.05. Participants with higher Healthy Lifestyle and Personal Control scores reported greater life satisfaction, *r*(462) = 0.27, *p* = 0.000. Healthy lifestyle alone predicted 8.8% of the variance in life satisfaction (*R*^2^ = 0.088, df 1462, *p* = 0.005). SEM confirmed the model had goodness-of-fit (χ^2^ = 2.63, *p* = 0.453). The satisfaction with life of dietitians is directly and positively influenced by a greater wellness orientation and personal healthy lifestyle practices. The effect of practice and lifestyle on life satisfaction appears to be greater for men.

## 1. Introduction

Health as defined by the World Health Organization is “a complete state of physical, mental and social well-being, and not merely the absence of disease or infirmity” [1]. With the emergence of personalized medicine based on genetic and epigenetic data, the health potential of an individual is partially determined by biology, but influenced by the environment and one’s personal wellness resources acquired throughout the lifespan [2]. Integrative health is a broader, more contextual construct that views wellbeing as not only physical, but involves an individual’s mind and spirit [3]. The term “health” does not appear in the “AND Definition of Terms”, a publication updated every five years, which demonstrates either a lack of shared meaning within the profession to this essential concept or the assumption that personal and professional philosophy are synonymous. Limited evidence on the wellness capacity of dietitians leaves a gap in our understanding of how dietitians’ health philosophy and lifestyle fit with their professional self-concept. Exploration of the Registered Dietitian Nutritionists’ core health values may improve the profession’s ability to address gaps in our understanding of health beliefs and behaviors, and their relationship to well-being [4].

The framework chosen for this study categorizes health philosophy as having clinical, functional, adaptive, and eudemonistic (wellness) components [5]. The Laffrey’s Health Conception Scale (LHCS) was developed to capture these dimensions, and subsequent studies have reduced scale to two stable factors: clinical and wellness [6,7,8]. The term “satisfaction with life” as measured by the Satisfaction with Life scale encompasses a global assessment of well-being and attainment of life goals [9]. Levels of life satisfaction are related to health status, health-related quality of life, health behaviors, social skills, engagement with life, and work [10]. The a priori hypotheses stated that there would be a positive correlation between one’s definition of health as health-promoting and the following variables: personal healthy lifestyle behaviors, integrative professional practices, and satisfaction with life.

In summary, the intent in undertaking this study was to explore any associations between health philosophy, lifestyle, professional practices, and satisfaction with life in a random sample of registered dietitian nutritionists (RDN).

## 2. Materials and Methods

A cross-sectional study of US dietitians using an online survey was planned. Following ethics approval of exempt status by the Rocky Mountain University of Health Professions Institutional Review Board, a random sample of 5000 dietitians currently registered with the Commission for Dietetic Registration was recruited through an email solicitation. For this study, the objective was to explore key issues and variables that may impact the satisfaction of life in RDNs. The effect size of 0.50 was chosen for a medium to large importance of effect [11,12]. An alpha level represents the odds that the observed result is due to chance and accepting 0.05 as the risk is acceptable for exploratory studies [13]. Using G*Power 3 software, the minimum sample size to detect a difference between two independent group means is 176 at power = 0.95 (1-β err prob) [14,15]. The web-based survey was administered using the Qualtrics© commercial software application during September 2016.

The questionnaire was comprised of four reliable and validated scales. Published estimates of Cronbach’s coefficient alpha internal reliabilities for the scales ranged from 0.72 to 0.95 [9,16,17,18]. The reduced version of Laffrey’s Health Conception Scale (LHCS) scale was used to identify the personal definition of health or health philosophy [8]. The Satisfaction with Life scale (SWLS) was the outcome variable of interest measuring individual perceptions of the meaning of health [19]. Separate papers were devoted to the validation process of the English version of the Healthy Lifestyle and Personal Control Questionnaire (HLPCQ) and Integrative Medicine (IM-30) [20,21] The HLPCQ measures personal routines and the degree to which the individual is empowered through lifestyle and health choices [17]. The IM-30 measures professional practice patterns along a continuum from traditional to integrative styles of practice for physicians. Some items were eliminated from the original instrument and those retained had the best fit with the scope of practice of the RDN. The survey included demographics and role identities identified through practice group memberships, licenses, or certifications.

An email was ultimately delivered to 4561 addresses from the initial list. Some emails with active addresses were undeliverable due to either firewall restrictions or spam filters. After reading study information about their rights as a participant, respondents were given the opportunity to participate or opt-out of the study. Four dietitians chose to “opt-out”, while 129 others “opted-in” but terminated participation without answering any questions. The questionnaire took an average of 15 min to complete. Following the initial solicitation; reminders were sent to non-responders seven days following the first reminder; and two weeks later. The survey was closed to participation approximately four weeks after it began. The overall response rate was 11.4%, with a completion rate of 76%. The characteristics of the survey participants are in Table 1.

Data were analyzed using SPSS^®^ version 22.0 (IBM, Armonk, NY, USA); statistical significance was defined as *p* < 0.05 [22]. Data was screened for accuracy, normality, completeness and homogeneity of variance. Survey participation was reviewed for opt-in, opt-out, non-completers, and those included in the final analysis. The outlier labeling technique identified cases that fell outside the interquartile range ±2.2 × (Q3–Q1) and these cases with a mean life satisfaction of less than 3.2 were filtered from the analyses. There were 2 cases with missing responses for LS and 25 for HPLCQ, therefore these cases were removed. The Kaiser-Meyer Olkin (KMO) statistic and Bartlett’s Test of Spherocity (KMO = 0.880, χ^2^ = 6060.6, *p* < 0.000) of SWLS were used to determine if sample size was suitable for the confirmatory factor analysis (CFA). For LHCS and IM-26 responses, a group mean was imputed for missing values. The flowchart of recruitment and selection is presented in Figure 1.

Descriptive data was examined and compared to demographics published by the Commission on Dietetic Registration. Participation reflected the general population of dietitians insuring generalizability of results. Dataset is available at https://data.mendeley.com/datasets/nxb2ybyj4x/1 (also included in Appendix A).

## 3. Results

A factor analysis was performed on each scale, and reliabilities were inspected. Normative data from multiple sources were compared (Table 2). Coefficient alphas for each scale were similar to those reported previously in adult samples as shown in Table 3 [8,16,17,19].

Internal consistency, and reliabilities of the scales are also presented in Table 3 [16,17,23,24]. The life satisfaction scale consisted of 5 items (α = 0.900), full health conception scale consisted of 16 items (α = 0.900), healthy lifestyle and control scale consisted of 26 items (α = 0.812), the integrative medicine partial scale had 25 items (α = 0.886). The reduced LHCS scale consists of two subscales that were used in separate analyses: clinical (α = 0.914) and wellness (α = 0.894). Study data also supports a three-factor health conception, with the unique loadings for functional beliefs as a separate component as in earlier studies. A few of the HPLCQ instrument items, however, did not load on the same factors as reported by Darviri and associates [17,20]. Regardless, the analyses in this paper did not require use of the individual factor components. The responses to the reduced LHCS, IM, and HLPCQ, and SWLS were summarized into index scores, and analyzed as continuous measures.

Pearson correlation coefficients between the index scores and demographic variables were analyzed to determine the degree of relationship between of the latent variables. A correlational analysis was conducted to examine the relationship between life satisfaction and the full health conception scale index, and the component subscales as shown in Table 4. The analysis included evaluating at the data for presence of general relationships and their directionality. An analysis of variance was performed to explore the significance of group differences and interactions on satisfaction with life, healthy lifestyle and personal control, integrativeness, and wellness philosophy. Potential mediators were explored and included workplace setting and years of practice as they related to previous studies or life satisfaction [16,25]. The correlation between related concepts, such as integrative practice and a wellness philosophy was weak and insignificant, yet larger than its relationship with clinical health philosophy. This suggests that the practice ideology of the US dietitian is in transition from traditional allopathic medicine to more holistic views of health.

Lastly, the literature suggests that life satisfaction is associated with behaviors that are health-promoting versus health-protecting. Our data supports an association between a comprehensive healthy lifestyle as measured by the HPLCQ and life satisfaction, (*r*(479) = 0.285, *p* = 0.000). As predicted there was a small association with dietary risk avoidance, such as avoiding soft drinks and fast food, and satisfaction with life (*r*(479) = 0.071, *p* = 0.05). The results are consistent with the view of professional dietitians that avoidance of artificial sweeteners and undesirable food chemicals are behaviors consistent with a healthier diet. Two additional items which loaded separately from the original validation study from “dietary healthy choices” could be interpreted as “diet quality”. The proactive behaviors of “eating a good breakfast” and “eating whole grain products” contribute to the nutrient density of the diet [26,27]. The health-promoting diet quality concept was also associated with life satisfaction (*r*(479) = 0.100, *p* = 0.05).

A review of the outlier group data shows that not only were the SWLS responses significantly lower (*p* = 0.000), but the HPLCQ index was lower than cases retained for analysis (Table 5). The predictors of life satisfaction were tested with structural equation modeling using SPSS^®^ Amos™ 23 (IBM, Chicago, IL, USA) to refine and confirm the model [28]. Structural equation modeling requires data with no missing values, therefore cases with missing data were imputed with the mean values for the SWL, IM-25 and HPLCQ indices. A Multiple-Groups model was tested by gender, and the model fit comparison showed no significant differences between genders (χ^2^ = 4.230, *p* = 0.121). In this small sample of 16 men, the effect of lifestyle and integrative practice on satisfaction with life, merits a future study because the effect of lifestyle on satisfaction (*r* = 0.694, *p* < 0.01), and practice style on lifestyle was stronger for men (*r* = 0.562, *p* < 0.001). The final model of the determinants of life satisfaction is shown in Figure 2 using only the data from female dietitians. The model fit for the data was acceptable on all fit indices (CFI = 1.00, TLI = 1.016, RMSEA = 0.000 (90% confidence interval: 0.000 < RMSEA < 0.075)). The chi-square test was small and non-significant (χ^2^ = 2.626, *p* = 0.453) indicating that the model demonstrated overall goodness of fit. The standardized estimates show that lifestyle has the greatest impact on satisfaction with life (*r* = 0.27, *p* < 0.05), and that practicing in a more integrative way impacts favorably in a dietitian’s lifestyle practices (*r* = 0.13, *p* < 0.05). On the other hand, wellness philosophy as a predictor had a smaller effect on satisfaction with life (*r* = 0.10, *p* < 0.05).

## 4. Discussion

The aim of the study was to find evidence supporting three hypotheses. First, we posited that the health-promoting concept of health, or the wellness component of health philosophy, would be positively and significantly associated with life satisfaction. The full LHCS scale was not significantly associated with life satisfaction. As predicted, having a health philosophy that favors wellness was positively and significantly associated with life satisfaction, while the clinical health concept had a non-significant negative association. The finding that there is a stronger impact of lifestyle and integrative practice on male satisfaction with life deserves future investigation with a larger male sample.

The practice of complementary and integrative medicine is more functional and wellness oriented, while biomedical practitioners are more clinically oriented. The second hypothesis posited that the health-promoting concept of health would be positively associated with integrative medicine practice orientation. This was partially supported and analyzed in two ways. First, the health conception index score was not significantly associated with the overall integrative medicine index score. In the second analysis only one component of integrative medicine orientation, the complementary and alternative knowledge factor, was positively and significantly associated with the health conception-wellness factor. Admittedly, the effect size was small, but in a separate paper the authors demonstrated that integrative nutrition practice diffusion is in the early phase of adoption, but is more common within certain communities of practice. A separate paper described the characteristics and evidence-based professional certifications and licenses earned by dietitians in the specialization of integrative and functional medicine [21].

It is not surprising to find that individuals who are less satisfied with their lives report one or more wellness dimensions as less than optimal in their daily lives. This finding supports the prediction that healthy lifestyles support well-being. It also suggests that dietitians who do not live a wellness lifestyle, a situation which is in conflict with their role identity, feel less satisfied with their lives.

## 5. Conclusions

The aim of this study was to explore the health conception or philosophy of Registered Dietitian Nutritionists and to see if it was related to satisfaction in life. Wellness-oriented philosophy, integrative medicine practices, and healthy lifestyle were hypothesized to be positively associated with life satisfaction, but there was no prediction of the influence of gender. One implication of these findings is that the pre-professional curriculum should focus on wellness knowledge and methods of personal health self-management. Dietetic educators should use nutritional and assessment labs to set baseline goals for the undergraduate student and use digital technology to track progress until graduation. A simulation lab could be designed to predict wellness trajectories based on current fitness levels to revise routines and fitness goals in a dynamic fashion. Our data suggests that practicing dietitians perceive greater satisfaction with their lives if they have healthy lifestyles gained from making healthy diet choices, physical activity, and seeking social-emotional understanding from family and friends a daily practice. Our analysis of the outliers found that for some dietitians, they have not developed healthy lifestyles. Since this impacts their wellbeing, allied health educators should include healthy self-development courses and community service projects into the curriculum at the college and post-graduate levels.

The finding that men in dietetics are less satisfied in life than their female counterparts is disconcerting. We did not expect that a significant relationship would be detected as men make up a small percentage of the dietetic workforce, approximately 3.4%, but our male response rate matched the actual ratio of males to females who are US RDNs [29]. Exploration of the essential sources for their perception should be performed with a qualitative narrative technique, but our findings suggest that male dietitians are less likely to practice lifestyle behaviors as a route to optimal health, and more likely to use risk-avoidance for this purpose.

Minority status of men within dietetics, as it was for nursing, may have consequences in the classroom and in the clinical setting [30]. Findings from a professional nursing self-concept study showed little overlap between wellness values and the ability for self-care [31]. Instead, nurses were more likely to personally value spiritual growth and interpersonal relationships, rather than nutrition conscious and physically active lifestyles. Our instrument did not include questions on spirituality, but the interpersonal relationship responses show that that is valued among dietitians, as well. Although men make up 36% of physical therapists, data from one small study women demonstrated an advantage in their exercise frequency and greater Aerobic Capacity Wellness [32,33]. Future research should explore gender differences in wellness value development throughout the lifespan, and professional barriers to a healthy lifestyle.

## Figures and Tables

**Figure 1 behavsci-07-00067-f001:**
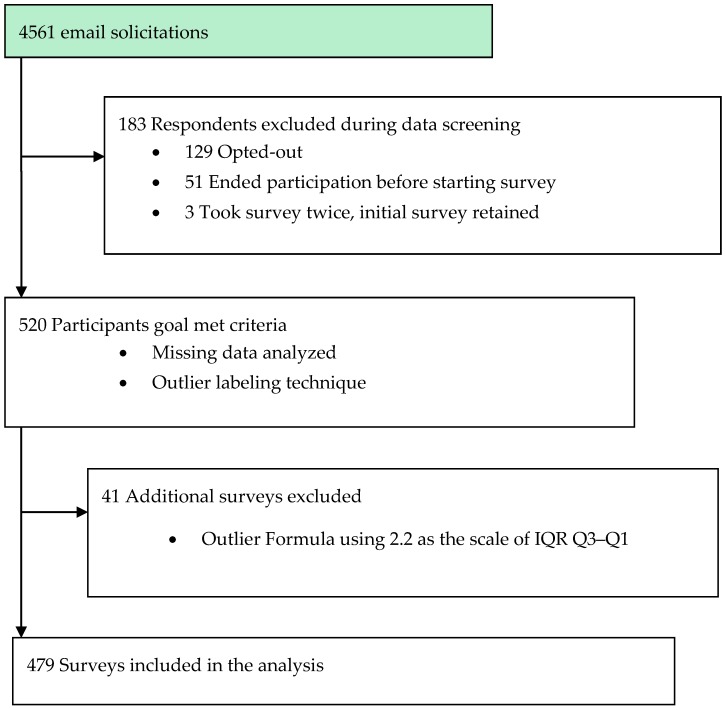
Flowchart of participant recruitment and case selection process.

**Figure 2 behavsci-07-00067-f002:**
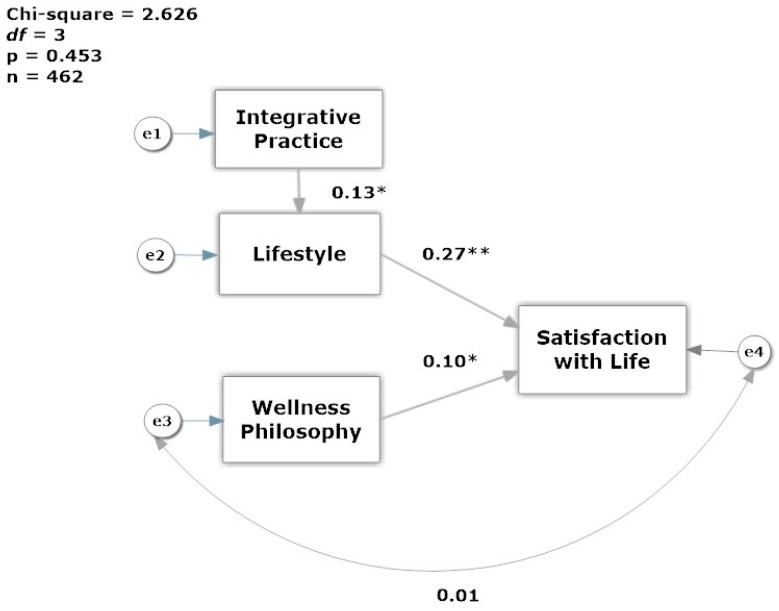
Model of Life Satisfaction of Registered Dietitian Nutritionists. Note: Standardized regression weights shown; Integrative Practice = IM-26, Lifestyle = HPLCQ, Wellness Philosophy = LHCS, and SWLS = Satisfaction with Life. * *p* < 0.05, ** *p* < 0.001.

**Table 1 behavsci-07-00067-t001:** Demographic characteristics of Registered Dietitian Nutritionists.

	N	Percent
Gender	479	100
Female	462	96.5
Male	16	3.3
Education	476	99.4
BS	208	43.4
MS	241	50.3
PhD	27	5.6
Years in Practice	479	100
Less than 1	25	5.2
1–4	96	20.0
5–9	81	16.9
10–14	48	10.0
15–25	76	15.9
25–34	98	20.5
Greater than 35	55	11.5
Primary Role	479	100
Clinical	179	37.4
Community	208	43.4
Education	45	9.4
Management	23	4.8
Outside Dietetics	20	4.2
Unemployed/Retire	4	0.8

**Table 2 behavsci-07-00067-t002:** Normative data for satisfaction with life, health conception, healthy lifestyle, and integrative medicine scales.

Scale/Sample	N	Mean Score (95%CI)	SD	Range
**Life Satisfaction**				
Dietitian Study	477	27.12 (26.7–27.5)	4.37	17–35
Adult Women ^a^	171	22.90	6.7	5–35
**Health Conception ***				
Dietitian Study	477	69.85 (64.7–66.7)	11.66	22–96
Health Conception-Clinical		24.28 (23.6–25.0)	7.87	7–42
Health Conception-Wellness		41.33 (40.9–41.9)	5.66	9–54
Adult Women ^b^	90			
Health Conception-Clinical		35.82	8.08	18–48
Health Conception-Wellness		38.94	5.90	23–48
**Healthy Lifestyle and Personal Control**				
Dietitian Study	479	73.50 (72.9–74.2)	8.80	42–97
Adult Sample (Greece) ^c^	285	64.61	11.84	35–98
**Integrative Medicine ^†^**				
Dietitian Study	440	71.23 (67.2–69.5)	12.98	39–108
Physicians, Chiropractors, and Accupuncturists ^d^	294	62.25	18.75	43–74

*Note*: * Reduced Lafferty Health Conception Scale with 16 items only. ^†^ IM-26 scale items. Data from ^a^ Judge, 1990 [24], ^b^ Gasalberti, 1999 [23], ^c^ Darviri, et al., 2014 [17], ^d^ Hsaio, et al., 2005 [16].

**Table 3 behavsci-07-00067-t003:** Descriptive statistics and internal consistency of Satisfaction with Life (SWL), Health Conception (HC), Healthy Lifestyle and Personal Control Questionnaire (HPLCQ) and Integrative Practice (IM)-26 scales and subscales.

Scale/Domain	No. of Items	Mean Score (SD)	Item Means	Internal Consistency ^†^
Life Satisfaction	5	27.12 (4.37)	5.42	0.809
Health Conception	16	69.85 (11.66)	4.11	0.900
Clinical	7	41.33 (5.66)	3.47	0.914
Wellness	9	24.28 (7.87)	4.60	0.894
HPLCQ	26	73.5 (8.80)	2.83	0.812
Integrative Medicine-26 *	26	71.23 (12.98)	2.74	0.886

*Note*: * Possible range 0–80, with higher scores indicative of greater Integrative Medicine orientation. ^†^ Cronbach’s coefficient alpha.

**Table 4 behavsci-07-00067-t004:** Pearson’s product moment correlations for life satisfaction.

Variables	Satisfaction	HC Index	HC Wellness	HC Clinical	IM-26	HPLCQ	Gender
Satisfaction	1						
HC Index	0.038	1					
HC-Wellness	0.206 **	0.514 **	1				
HC-Clinical	−0.013	0.860 **	0.228 **	1			
IM-26	0.079	−0.010	0.066	−0.032	1		
HPLCQ	0.357 **	0.003	0.093 *	0.074	−0.006	1	
Gender	−0.144 **	−0.141	−0.133 **	−0.003	0.079	0.001	1

*Note*: ** correlation is significant at the 0.001 level (two-tailed); * correlation is significant at the 0.05 level (two-tailed), *n* = 497. Satisfaction, IM-26, and HC (Health Conception) Index are summated scale scores. HC Wellness and HC Clinical are subscales within HC Index.

**Table 5 behavsci-07-00067-t005:** Comparison of outliers to analysis group.

Scale/Sample	N	Mean Score	SD	Range	Skewness
**Life Satisfaction-Full Scale**					
Analysis Group	477	27.12	4.37	17–35	−0.29
Outliers	41	12.85	2.46	6–16	1.813 **
**Health Conception-Full and Subscales ^a^**					
Analysis Group	448	65.70	11.52	22–96	−0.73
Outliers-	39	65.00	10.68	30–80	−1.23
Clinical Subscale	463	24.31	8.00	7–42	−0.33
Outliers	41	24.61	6.61	9–35	−0.48
Wellness Subscale	461	41.39	5.76	9–54	−1.86
Outliers	39	39.94	0.70	14–49	−1.97
**Healthy Lifestyle and Personal Control-Full Scale**					
Analysis Group	454	73.50	8.80	42–97	−0.73
Outliers	39	64.87	9.74	45–81	−1.23 **
**Integrative Medicine-26 ^b^**					
Analysis Group	440	68.38	13.46	36–107	−0.015
Outliers	36	63.80	11.75	36–81	−0.554

*Note*: ** = *p* < 0.001. ^a^ Laffrey Health Conception—Reduced Scale, 16 items only. ^b^ IM-26 reduced scale, 26 items.

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
