# Peer review of "Health Philosophy of Dietitians and Its Implications for Life Satisfaction: An Exploratory Study"

_behavsci, 2017, doi:10.3390/bs7040067_

Round 1

Reviewer 1 Report

Lines 229 to 242:

An N of 3 males makes me question the relevance any results, discussion, suggestions or conclusions re male RDNs, except to mention  how few there are.

Since this study did not even mention RDN values re spiritual growth and interpersonal relationships it seems totally out of context to compare nurses or males RDNs in this paragraph...Interesting topic and related to personal life satisfaction,, but not relevant here. 

Double check: line 87:  information ON?     line 161:  analysis WILL?

Table 5:Table would make comparisons easier if Health Conception section was rearranged so that outlier data was just beneath the Analysis Group data for each individual item.  

I was shocked to read that the term "health" was not included in the "AND Definition of Terms" 

I was glad to see the Outliers addressed. 

Author Response

Lines 229 to 242:

An N of 3 males makes me question the relevance any results, discussion, suggestions or conclusions re male RDNs, except to mention how few there are.

There were 16 males who comprise 3.3% of the participants which matches the national registration ratio of male to females.

Since this study did not even mention RDN values re spiritual growth and interpersonal relationships it seems totally out of context to compare nurses or males RDNs in this paragraph...Interesting topic and related to personal life satisfaction, but not relevant here.

A reviewer for a different journal suggested that I should have compared the lifestyle behaviors dietitians  to a sample of adults with chronic illnesses. The instrument was designed for healthy adults. My point is suggesting that this lifestyle questionnaire should be modified for use with a chronically ill population. I have revised this section because the lifestyle instrument did address interpersonal relationships and routine communications.

Double check: line 87:  information ON?     line 161:  analysis WILL?

Yes, I changed the wording in both lines. Thank you.

Table 5:Table would make comparisons easier if Health Conception section was rearranged so that outlier data was just beneath the Analysis Group data for each individual item. 

The table was revised as you suggested.

I was shocked to read that the term "health" was not included in the "AND Definition of Terms"

I am no longer shocked because disease and risk factors are strong components of the curriculum. I use the health conception instrument to stimulate discussion on health philosophy in my graduate level community nutrition course. Sadly, recent graduates of dietetic internships have never reflected on the outcomes they want to achieve with their patients.

I was glad to see the Outliers addressed.

I rarely see an outlier analysis and felt that they would tell a story, more than just a percentage of people windsorized out of the analysis. 

Reviewer 2 Report

Thank you for allowing me to read this paper.  It is well written and interesting. I think it would be very specific to the USA system of education and practice, and it would be interesting to compare with dietitians in another country.  I found the comment that you were surprised by the male results interesting.  I understood that men have a different approach to healthy living compared with women, and I would have thought that these results might mirror men in other fields of endeavour.  I was interested in your comments that the philosophy of a health promoting diet was common in US dietitian nutritionists, while the prevalence of obesity is still high.  How do you account for this?  Is it the outliers who have more struggles with their weight for example.  I think the results are interesting to the profession, but I am not certain what the implications for practice or educational programs is, in either dietetics or allied health professions generally and this could be strengthened in the conclusion. I was unclear what "pre-professions curriculum should focus on wellness knowledge and methods of personal health self- management" could mean in practice.  Could you give some kind of example, so that those in other parts of the world can gain more understanding? The data on outliers is interesting - are you suggesting that these professionals are less effective or simply less satisfied and does that actually matter?

Author Response

Thank you for allowing me to read this paper.  It is well written and interesting. I think it would be very specific to the USA system of education and practice, and it would be interesting to compare with dietitians in another country.  I found the comment that the male results interesting surprised you.  I understood that men have a different approach to healthy living compared with women, and I would have thought that these results might mirror men in other fields of endeavour.  I was interested in your comments that the philosophy of a health promoting diet was common in US dietitian nutritionists, while the prevalence of obesity is still high.  How do you account for this?

The mean for the responses to the diet questions was high, but much lower for physical activity. Studies have shown that PA is not as important for weight loss, but critical for healthy weight maintenance. We need both approaches to overall health and longevity. Content knowledge has a small, but important effect on health behavior, but daily lifestyle practice has a big effect on wellbeing. One’s academic years may facilitate exercise, but career, commute, and family obligations may take time away from self-care.

Is it the outliers who have more struggles with their weight for example. 

That may well be true, we did not ask any weight related questions. The principal author published a survey study on weight history parents and attitudes toward childhood obesity. It appears that stigma may play a role for parents ignoring health providers and media warnings.

I think the results are interesting to the profession, but I am not certain what the implications for practice or educational programs is, in either dietetics or allied health professions generally and this could be strengthened in the conclusion. I was unclear what "pre-professions curriculum should focus on wellness knowledge and methods of personal health self- management" could mean in practice.

The dietetics curriculum has requirements that include nutrition, basic science, management, statistics, and nutrition counseling. There are no requirements for kinesiology oriented courses or community service projects on physical activity. How can dietitians counsel clients if they do not understand the theory and literature in health and wellness, not just disease?

 Could you give some kind of example, so that those in other parts of the world can gain more understanding?

I gave the example of the use of nutritional assessment and fitness testing laboratories and simulations as health & fitness development for the student. Thank you for that suggestion.

The data on outliers is interesting - are you suggesting that these professionals are less effective or simply less satisfied and does that actually matter?

Men in dietetics tend to be more physical activity minded, and many go into sports nutrition. That is why I was surprised that they did not have healthier lifestyles. But male nurses also have wellness deficiencies, so minority status in their profession may impact their satisfaction with life. It may be a sense that the lifestyle that they value and promote in practice is difficult to manage in their lives.  I am on the dissertation  committee for a physical therapist in health sciences. She is using the same instrument for lifestyle and health philosophy with a sample of physical therapists. Those findings should be interesting. There is some data to suggest that physical activity among physical therapists decline after they enter practice. Perhaps we do not prepare them for lifestyle change to address inactivity and stress from working long hours.

Weight gain in the United States is due to many factors. The dietitians in my sample did not have a high level of physical activity. Regretfully, our  professional organization is still promoting processed foods “in moderation” and accept money from food and beverage companies. This sends a confusing message to members and the public. I also study the microbiome and its effects on health. The US lifestyle and food environment contributes to the development of dysbiosis. Data suggest that emulsifiers and antibiotics are deleterious to our gut bacteria, and the pathogenic survivors are known to promote obesity. I have seen slender foreign graduate students gain more than 20 pounds in their first semester. Those students who practice whole foods, plant-based diets fare somewhat better.

The outliers have a much lower level of satisfaction with their lives. Dietetics is a low paying profession with high standards. The survey did not contain a question about salary.

For some dietitians, their lifestyle behaviors were problematic, but they had a wellness health philosophy. I wonder if they were distressed by the disconnect between what they believe about health behaviors and how they live.

We are creating a new undergraduate track in our nutrition department - Nutrition and Fitness. We need more dietitians trained and active in fitness management in the United States. 

Reviewer 3 Report

In the first sentence of the introduction, do you mean "World Health Association" or World Health Organization"?

The methods should start with design and setting not the questionnaire. 

The authors mention that their sample was random.  How? From which sampling universe / sampling frame? What type of random? More explanation and details are needed. 

We do not need to see data on outliers. 

Errors are only required for endogenous variables (those with an incoming paths). It is not clear why gender and other exogenous variables have errors.  Nothing predicts gender. Gender does not need error, only needs covariance.... 

The discussion does not need exact numbers. That is one of the many differences between a results and a discussion section. 

Please discuss some of the weak correlations in the correlation matrix. Why some of the correction between similar constructs are weak?

The study has not included age, years in practice and practice setting to the correlation table. That would help us understand how stable these measures are...

I did not find data on consent. This is very essential.

Author Response

In the first sentence of the introduction, do you mean "World Health Association" or World Health Organization"?

The definition is from the constitution of the World Health Organization, thank you for the correction.

The methods should start with design and setting not the questionnaire.

I have made that change to the manuscript.

The authors mention that their sample was random.  How? From which sampling universe / sampling frame? What type of random? More explanation and details are needed.

The Commission on Dietetic Registration took a random sample from the sampling universe of 89,300 registered dietitians. I have requested more information on methodology from their administrator.

We do not need to see data on outliers.

The other reviewers disagree. But one did request a different table arrangement for comparison.

Errors are only required for endogenous variables (those with an incoming paths). It is not clear why gender and other exogenous variables have errors.  Nothing predicts gender. Gender does not need error, only needs covariance....

Thank you for pointing out this problem.  When using a single indicator, like gender, the researcher must assume the item is measured without error. I tried to constrain the gender variable with the assumption that there is “0” error variance, but the model was not supported by the data. I then tried a Multiple Group analysis by gender and found that the model had good fit. There were minor differences between groups, but both models had good fit. Likely, the small number of men (16) makes and differences trivial and under-powered with the male sample size. I would not have seen this without using the Multiple Group comparison, and I appreciate your input. I analyzed them separately and found significant differences. I discussed the need for a larger sample of men to determine if this difference is a valid finding.

The discussion does not need exact numbers. That is one of the many differences between a results and a discussion section.

I will revise the discussion section as you recommend.

Please discuss some of the weak correlations in the correlation matrix. Why some of the correction between similar constructs are weak?

I will revise as you suggest.

The study has not included age, years in practice and practice setting to the correlation table. That would help us understand how stable these measures are...

The age demographics were categorical ranges, but we converted them to ordinal. The only significant bivariate correlation was between gender and years in practice. Practice settings were not binary, they were descriptive and we have previously published that data. We have referenced that article.

I did not find data on consent. This is very essential.

I will add them. 

Round 2

Reviewer 3 Report

The author has appropriately responded to the comments and concerns raised by reviewers. As a result, the paper is now improved in many aspects. I appreciate the opportunity to review this paper. I recommend publication.